# Pulmonary Biodistribution of Platelet-Derived Regenerative Exosomes in a Porcine Model

**DOI:** 10.3390/ijms25052642

**Published:** 2024-02-24

**Authors:** Skylar A. Rizzo, Monique S. Bagwell, Paige Schiebel, Tyler J. Rolland, Ryan C. Mahlberg, Tyra A. Witt, Mary E. Nagel, Paul G. Stalboerger, Atta Behfar

**Affiliations:** 1Van Cleve Cardiac Regenerative Medicine Program, Mayo Clinic Center for Regenerative Medicine, Rochester, MN 55905, USA; 2Mayo Clinic Medical Scientist Training Program, Rochester, MN 55905, USA; 3Mayo Clinic Department of Molecular Pharmacology and Experimental Therapeutics, Mayo Clinic, Rochester, MN 55905, USA; 4Mayo Clinic Department of Cardiovascular Medicine, Rochester, MN 55905, USA; 5Marriott Heart Disease Research Program, Mayo Clinic, Rochester, MN 55905, USA

**Keywords:** extracellular vesicles, exosomes, biodistribution, pulmonary

## Abstract

The purpose of this study was to evaluate the biodistribution of a platelet-derived exosome product (PEP), previously shown to promote regeneration in the setting of wound healing, in a porcine model delivered through various approaches. Exosomes were labeled with DiR far-red lipophilic dye to track and quantify exosomes in tissue, following delivery via intravenous, pulmonary artery balloon catheter, or nebulization in sus scrofa domestic pigs. Following euthanasia, far-red dye was detected by Xenogen IVUS imaging, while exosomal protein CD63 was detected by Western blot and immunohistochemistry. Nebulization and intravenous delivery both resulted in global uptake of exosomes within the lung parenchyma. However, nebulization resulted in the greatest degree of exosome uptake. Pulmonary artery balloon catheter-guided delivery provided the further ability to localize pulmonary delivery. No off-target absorption was noted in the heart, spleen, or kidney. However, the liver demonstrated uptake primarily in nebulization-treated animals. Nebulization also resulted in uptake in the trachea, without significant absorption in the esophagus. Overall, this study demonstrated the feasibility of pulmonary delivery of exosomes using nebulization or intravenous infusion to accomplish global delivery or pulmonary artery balloon catheter-guided delivery for localized delivery.

## 1. Introduction

Extracellular vesicles are membrane encapsulated structures secreted by nearly all cell types [1]. Subtypes of extracellular vesicles include exosomes, microvesicles, and apoptotic bodies, which are grouped based on size and biosynthesis [1]. Exosomes in particular have various functions in cell-to-cell communication and have been widely studied for their potential as biomarkers, their possible regenerative applications, or their use as therapeutic delivery vehicles [2]. However, effective delivery methods and biodistribution of exosomes are still under investigation. The pharmacokinetic behavior of exosomes has primarily been evaluated in mice, where systemic delivery results in extracellular vesicle uptake in the liver, spleen, gastrointestinal tract, and lungs [3]. To target delivery, methods have primarily focused on modifications of exosome surface proteins, either by modification of the exosome or of the cell producing the exosome, to allow for specific uptake in the tissue of interest [4]. 

Exosomes, particularly mesenchymal stem cell-derived exosomes, have been increasingly investigated for the treatment of pulmonary diseases, such as acute respiratory distress syndrome, chronic obstructive pulmonary disease, asthma, and pulmonary fibrosis [5]. The therapeutic impact of mesenchymal stem cell-derived exosomes is largely believed to be related to anti-inflammatory, antioxidant, and anti-apoptotic effects [6]. Further, various types of exosomes have been employed as drug carriers for chemotherapeutic agents, facilitating selective delivery of the drug to cancer cells [7]. Noninvasive delivery approaches for exosomes such as nebulization and systemic intravenous delivery have been tested; however, the vast majority of biodistribution studies have been conducted in rodents [8,9]. However, biodistribution of extracellular vesicles can vary significantly between large and small animal models due to differences in fenestrae size causing different uptake patterns [10].

Previously, we have shown a platelet-derived exosome product (PEP) is capable of tissue regeneration in the context of wound healing [11,12,13,14]. These exosomes donate various proteins, including transforming growth factor beta, NF-κB, and programmed death ligand 1, which drive exosome effects on recipient cells [13,14]. Additionally, PEP can augment cell proliferation and angiogenesis both in vitro and in vivo [11,13]. Due to the immune-modulating, angiogenic, and proliferative effects of PEP, we hypothesize that platelet exosomes would have similar efficacy compared to mesenchymal stem cell-derived exosomes in treating lung disease.

The objective of this study was to ascertain the ideal procedural methodology for pulmonary delivery of platelet-derived exosomes in a porcine model. A large animal model and clinically feasible procedural approaches were utilized to maximize the translatability of the findings. 

## 2. Results

### 2.1. Exosome Characterization of Platelet-Derived Exosome Product (PEP) and Membrane Labeling by Lipophilic Dye

NanoSight analysis of PEP demonstrated a mean size of 136.3 nm (±standard error 0.9 nm) and total concentration 5.37 × 10^12^ particles/mL (±standard error 1.5 × 10^11^ particles/mL) (Figure 1A). The typical exosome size is considered to be 50–150 nm [15]. Transmission electron microscopy images of PEP demonstrated similarity in the size of exosomes in the sample population (Figure 1B). Western blot demonstrated the presence of exosomal marker proteins CD63, Flotillin, and CD9 [15], and also showed that platelet marker CD41 was preserved in platelet-derived exosomes (Figure 1C). Dialkylcarbocyanine dyes are lipophilic dyes used for exosome tracking that are minimally fluorescent until incorporated into lipid membranes [16]. Red fluorescent 1,1′-dioctadecyl-3,3,3′,3′-tetramethylindocarbocyanine perchlorate (DiI) and far-red fluorescent 1,1′-dioctadecyl-3,3,3′,3′-tetramethylindotricarbocyanine iodide (DiR) dyes were used in this study. DiI dye labeling was performed with PEP or sterile water (control) and incubated with human umbilical vein endothelial cells (HUVECs) for 30 min to demonstrate the cellular uptake of exosomes (Figure 1D). Fluorescent signal per cell was quantified and demonstrated a mean of 3349 (±2197 SD, *p* = 0.0079) compared to control (mean 227.6 ± 143.8 SD) (Figure 1E). To further evaluate the tracking utility of DiI and DiR, Xenogen IVUS imaging was employed. DiI demonstrated a greater relative signal in PEP labeled with DiI, but there was background signal in the sterile water labeled with DiI (DiI) (Figure 1F). However, DiR did not show any background in the sterile water labeled with DiR control (DiR) but did demonstrate signal in DiR-labeled PEP (Figure 1G). Therefore, the far-red DiR dye was selected for further use in vivo to track the delivery of PEP.

### 2.2. Pulmonary Exosome Delivery Procedures

To achieve targeted pulmonary delivery of PEP, the following delivery methods were tested: nebulized, PA balloon catheter-guided, and intravenous. A cartoon graphic demonstrating these approaches is shown in Figure 2A. Nebulization was accomplished with a jet nebulizer attached to an endotracheal tube (Figure 2B). A radiographic image demonstrating the endotracheal tube position above the carina is shown (Figure 2C). To facilitate the PA balloon catheter-guided delivery, pulmonary angiograms were acquired (Figure 2D). The balloon catheter was then used to selectively enter a single branch of the pulmonary artery. The balloon was wedged to occlude forward blood flow, demonstrated with contrast injection (Figure 2E,F). Blood flow occlusion with PEP delivery in the pulmonary artery branch for 5 min allowed for exosome absorption. Intravenous PEP delivery was performed through the femoral vein.

### 2.3. Biodistribution of PEP in Lung Tissue

Following PEP delivery by intravenous, PA balloon catheter-guided, or nebulized approaches, lung tissue was removed and imaged with Xenogen IVUS to track far-red DiR dye. Intravenous and nebulized methods achieved global lung delivery, while the PA balloon catheter-guided approach was able to localize delivery to a single area (Figure 3A). CD63 was employed to detect PEP in pig tissue, since CD41, Flotillin, and CD9 all demonstrated significant cross-reactivity with control pig tissue. Control tissues were isolated from pigs that were not treated with PEP. Western blot was used to demonstrate the presence of exosomal protein CD63 in lung tissue (Figure 3B). Quantification of CD63 normalized to actin demonstrated nebulization had the highest delivery, with a mean of 0.989 (±SD 0.15) compared to the level of control at 0.029 (Figure 3C). The PA balloon catheter-guided delivery resulted in a similarly high mean of 0.959 (±1.24 SD) with greater variation between animals (Figure 3C). Intravenous delivery resulted in the lowest mean CD63 signal at 0.649 (±0.47 SD) (Figure 3C).

### 2.4. Biodistribution of PEP in Off-Target Tissues

Uptake of PEP in off-target organs including the heart, liver, spleen, and kidney was similarly evaluated by Xenogen IVUS imaging. No uptake was noted in the liver, heart, spleen, or kidney as a result of any of the methods described (Figure 4A). To assess PEP tissue uptake at higher resolution, Western blot of CD63 was utilized. Indeed, the liver tissue showed variable presence of PEP (Figure 4B). Nebulization-treated animals showed the greatest amount of liver uptake with quantification of CD63 fluorescent signal normalized to GAPDH of 0.131 (±0.05 SD) compared to control liver tissue at 0.017, while intravenous (mean 0.034 ± 0.01 SD) and PA balloon catheter-guided (mean 0.021 ± 0.01 SD) animals had levels that were closer to those of the control tissue (Figure 4C). There was no CD63 protein detected in the heart, spleen, or kidney samples of PEP treated animals (Figure 4D–F). For nebulized animals, the esophagus and trachea were also evaluated. Xenogen IVUS imaging demonstrated no esophageal signal, but scattered signal in the trachea (Figure 5A). The endotracheal tube was also evaluated after nebulization in comparison to an endotracheal tube without exposure to nebulized PEP. After nebulization, there was PEP signal throughout the entire endotracheal tube (Figure 5B). Western blot confirmed the absence of PEP in the esophagus (Figure 5C) and, as expected, its presence in trachea (Figure 5D). Quantification demonstrated the mean CD63 signal normalized to GAPDH in the esophagus of nebulized animals (0.005 ± 0.003 SD) was similar to that of the control esophagus (mean 0.002), while the trachea showed greater amounts of CD63 (mean 0.062 ± 0.01 SD) compared to the control trachea at 0.030 (Figure 5E).

### 2.5. Histologic Analysis of PEP in Lung Tissue

To further demonstrate the presence of PEP in lung tissue and demonstrate the specific areas of tissue uptake, immunohistochemical staining was performed for CD63. Compared to control lung tissue which showed minimal staining, there was CD63 detected in animals treated by intravenous, PA balloon catheter-guided, and nebulization approaches (Figure 6A). Quantification demonstrated a significant increase in CD63 detected per total tissue area in nebulization (mean 0.283 ± 0.02 SEM, *p* = 0.0344), PA balloon catheter-guided (mean 0.196 ± 0.03 SEM, *p* = 0.0009), and intravenous delivery (mean 0.137 ± 0.02 SEM, *p* < 0.0001) compared to control lung (mean 0.026 ± 0.01 SEM) (Figure 6B). These results recapitulate the trend seen by Western blot detection with the greatest CD63 signal detected in nebulization, followed by PA balloon catheter-guided and, lastly, intravenous delivery.

## 3. Discussion

In this study, intravenous, PA balloon catheter-guided, and nebulization methods were evaluated for regenerative platelet-derived exosome (PEP) delivery in a porcine model. Platelet-derived exosomes were employed due to their ability to promote tissue regeneration, particularly in the context of wound healing in various tissue types [11,12,13,14]. Xenogen IVUS imaging showed global lung uptake with nebulization and intravenous delivery, while PA balloon catheter-guided delivery demonstrated the ability to localize exosomes to a single lung area. To specifically validate the presence of PEP in tissues, exosomal marker CD63 was employed. Western blot demonstrated the presence of CD63 in the lung and liver with absence in the heart, spleen, and kidney. In the lung, CD63 was greatest after nebulization, followed by PA balloon catheter-guided delivery, and lowest following intravenous delivery. The liver showed the greatest amount of CD63 with nebulization, while intravenous and PA balloon catheter-guided methods yielded much lower amounts, likely secondary to oral ingestion. To further demonstrate the presence of CD63 within lung tissue, immunohistochemistry was performed for CD63 and resulted in a trend that paralleled Western blot results. CD63 was utilized here as it was highly human-specific and did not cross-react with porcine tissues. Overall, these results showed that exosomes demonstrate an aero-geometry that is compatible with nebulization, resulting in superior global lung uptake, while PA balloon catheter-guided delivery demonstrated a novel approach to localizing exosomes to specific regions of the lung.

PA balloon catheter-guided delivery showed the most variability of the approaches with regard to amount of exosome delivered to the lungs between animals. This finding was likely the result of the different pulmonary branches treated in each animal leading to different overall areas of lung being treated, thereby creating diversity in the quantity of exosomes within a particular bed. In future applications, targeting the same branch of the pulmonary artery may lead to greater consistency of that treatment. 

The liver uptake seen primarily with nebulization, with little uptake with intravenous or PA balloon catheter-guided delivery, was discordant with the pattern typically seen in rodent studies with significant liver uptake following intravenous delivery. However, one notable difference between the treatments was that nebulization takes longer to administer (30–60 min) while intravenous and PA balloon catheter-guided treatments were administered over 5 min. The longer period for nebulization may have allowed greater amounts of exosomes to be absorbed, especially if some of the nebulized product was ingested into the gastrointestinal tract. Extracellular vesicle absorption from the blood has been shown to occur within an hour [17]. Future studies should evaluate different time periods following administration of exosomes to better understand this finding. 

These delivery methods may also be effective with other types of exosomes that are being investigated in the treatment of lung diseases. Global lung delivery using intravenous or nebulized exosomes would provide an avenue to deliver exosomes for conditions effecting the entire lung, such as acute respiratory distress syndrome, chronic obstructive pulmonary disease, asthma, or pulmonary fibrosis. Meanwhile, delivery to specific lobes of the lung with PA balloon catheter-guided delivery could be employed by conditions affecting specific lung regions, such as lung cancer. Mesenchymal stem cell-derived exosomes have shown efficacy in animal models of acute respiratory distress; although scalable production remains a challenge, making delivery optimization key [18]. HEK293T-derived EVs loaded with Wnt protein can induce epithelial cell regeneration in a murine model of emphysema [19]. Extracellular vesicles derived from various cell types carrying chemotherapeutics, miRNAs, or tumor antigens are already being developed and have shown efficacy against lung cancer [20]. Specific testing of these delivery methods with additional types of extracellular vesicles is needed to determine if the efficacy is similar to that of platelet-derived exosomes. Extracellular vesicles have been shown preferential uptake from the cell type they originated from (autologous uptake); however, other cells types can absorb them as well (heterologous uptake) [21,22]. Cellular uptake of extracellular vesicles is largely thought to be related to particle size and surface molecules [23,24]. However, a study testing exosomes derived from five separate cell types demonstrated similar uptake in mice by intravenous injection [25]. 

In summary, this study represents a novel large animal study evaluating intravenous, PA balloon catheter-guided, and nebulization delivery as procedural approaches for exosome targeting to the lungs. The global lung delivery was greatest with nebulization, while PA balloon catheter-guided delivery resulted in localized delivery to a single lung area. This later finding may suggest that balloon catheter-based delivery may offer an opportunity for targeted delivery of exosomes in any arterial bed. This study also reinforced that biodistribution studies for exosomes likely require large mammals, as rodent models may not fully elucidate the pattern seen in large animals and thereby may not represent the pharmacokinetics seen in humans. Overall, these results showed that platelet-derived regenerative exosomes have a high proclivity towards pulmonary uptake and are well suited for therapeutic applications both in global and targeted delivery.

## 4. Materials and Methods

### 4.1. NanoSight Analysis

PEP samples were diluted in PBS prior to use in a NanoSight analysis NS300 system. Quantification of particle number and distribution of particle size were performed using NanoSight NS300 (Malvern Panalytical, Malvern, UK).

### 4.2. Transmission Electron Microscopy

Exosome samples were fixed in Trump’s fixative at 4 °C overnight and then applied to a carbon-coated nickel grid. The nickel grid was stained with 2% uranyl acetate, air-dried, and visualized using a JEOL 1400 plus TEM transmission electron microscopy (JEOL Ltd., Tokyo, Japan).

### 4.3. Western Blot 

Samples were lysed in a detergent-based buffer (10 mM HEPES pH 7.4, containing 1% Triton X-100, 50 mM sodium pyrophosphate, 50 mM sodium fluoride, 50 mM sodium chloride, 5 mM EDTA, 5 mM EGTA, 100 µM sodium orthovanadate, and 1:100 protease inhibitor cocktail [Sigma-Aldrich, St. Louis, MO, USA, P8340]). Control tissue isolated from pigs of the same age and species that were not treated with PEP was employed as control tissue for Western blot analysis. Protein lysates were quantified by Pierce^TM^ BCA Protein Assay (ThermoFisher, Waltham, MA, USA, #23227) and then loaded into polyacrylamide gels (Invitrogen, Carlsbad, CA, USA). Semi-dry transfer was performed using Trans-Blot Turbo (Bio-Rad, Hercules, CA, USA) onto nitrocellulose membranes. Membranes were incubated for 1 h in Intercept (TBS) blocking buffer (LI-COR Biosciences, Lincoln, NE, USA), followed by overnight incubation at 4 °C with primary antibodies with 0.1% Tween. Primary antibodies included CD63 (1:1000, R&D Systems, Minneapolis, MN, USA, #MAB50482), Flotillin (1:1000, Abcam, Waltham, MA, USA, #ab133497), CD9 (1:1000, Cell Signaling Technology, Danvers, MA, USA, #13403S), CD41 (1:1000, NovusBiologicals, Centennial, CO, USA, #NBP1-84581), GAPDH (1:2000, Cell Signaling Technology, 2118S), and actin (1:1000, LI-COR, #926-42212). After subsequent 1 h incubation with IRDye anti-rabbit and anti-mouse secondary antibodies (LI-COR), membranes were imaged using Li-Cor CLx Imager.

### 4.4. Immunocytochemistry

HUVECs were treated with DiI-labeled PEP or DiI in sterile water (control) for 30 min. Cells were then washed in PBS and fixed in 4% paraformaldehyde at 4 °C overnight. Permeabilization was performed with 0.1% Triton in PBS for 3 min. Cells were blocked (5% bovine serum albumin, 2% normal donkey serum, 0.02% Triton in PBS) for 10 min. Phalloidin (ThermoFisher, A12379) was added for 45 min at 37 °C. Mountant containing DAPI (ThermoFisher, P36935) was used to apply coverslips. Imaging was performed on confocal LSM 780 (Carl Zeiss Microscopy, LLC, White Plains, New York, NY, USA).

### 4.5. Lipophilic Dye Exosome Membrane Labeling 

Exosomes were labeled with a lipophilic dye to facilitate preliminary evaluation of tissue uptake. Here, DiR and DiI were reconstituted in DMSO at concentrations of 50 µM or 25 µM, respectively. For in vitro labeling, DiI was added to PEP to a final concentration of 5 µM and incubated at 37 °C for 10–20 min. Excess dye was removed by centrifugation through 100 kDa Amicon filters at 4000× *g* for 15 min. For in vivo labeling, DiR and DiI were added to PEP to a final concentration of 50 µM and excess dye was removed by the same centrifugation procedure. 

### 4.6. Animal Studies

All protocols were approved by the Mayo Clinic Institutional Animal Care and Use Committee (IAUC A00005398-20), which is a facility accredited by the Association for Assessment and Accreditation of Laboratory Animal Care (AAALAC). Intravenous injection and nebulization were tested for global pulmonary targeting, while pulmonary artery (PA) balloon catheter-guided delivery was evaluated for local delivery to a subset of lung tissue. Three- to four-month-old Sus scrofa domesticus pigs (Manthei Hog Farm, LLC, Elk River, MN, USA) weighing between 36 and 44 kg were used for all studies. Two female and one male pig were used for the intravenous and PA balloon catheter-guided groups, while three female pigs were used for the nebulization group. 

Sedation was performed by intramuscular injection of Telazol (5 mg/kg)/Xylazine (1–2 mg/kg) for intubation followed by continuous inhaled isoflurane (1–3%) for the duration of the procedure. Femoral vein access to place a 9 French sheath was performed under ultrasound guidance for intravenous and PA balloon catheter-guided delivery groups. Intravenous treatment with PEP was administered directly through a femoral vein sheath over 5 min. For PA balloon catheter-guided treatment, Swan Ganz catheter was advanced from the femoral vein to the pulmonary artery and a pulmonary angiogram was performed. A Storq wire was used to direct the catheter to a branch vessel and a balloon was wedged. Contrast was injected with angiography to monitor for a leak behind the balloon. Once adequate occlusion was achieved, PEP was injected over 5 min. Nebulization was performed with a jet nebulizer attached to the endotracheal tube with positive end expiratory pressure of 5 cm H_2_O and a respiratory rate 14–15. All animals received 1.5 × 10^13^ exosomes labeled with DiR far-red dye. Following treatments, animals were maintained on anesthesia for 15 min then euthanized with necropsy to collect the heart, lungs, liver, spleen, kidney, esophagus, and trachea for Xenogen IVUS imaging. Following imaging, samples from each organ were fixed in formalin or flash frozen.

### 4.7. Xenogen IVUS Imaging

At necropsy, the heart, lung, spleen, kidneys, liver, esophagus, and trachea were removed from the animal and rinsed in saline. In the Xenogen IVUS system, excitation 745 nm and emission 800 nm were used for DiR. Excitation 535 nm and emission 580 nm were used for DiI. Auto exposure was used to set the exposure time for each image.

### 4.8. Histologic Analysis

Tissues were rinsed in saline and fixed in 10% formalin overnight, followed by paraffin embedding. Control tissue isolated from pigs of the same age and species that were not treated with PEP was employed as control tissue for the immunocytochemistry studies. Immunocytochemistry was performed with ImmPRESS^®^ HRP Horse Anti-Rabbit IgG Polymer Detection Kit (Vector Laboratories, Newark, CA, USA, #MP-7401) and ImmPACT^®^ DAB (Vector Laboratories, #SK-4105) kits. Briefly, the slides were deparaffinized by sequential washes in xylene, 100% ethanol, 95% ethanol, and distilled water. Antigen retrieval was performed with 10 mM sodium citrate buffer pH 6.0 with 0.05% Tween and heated in a pressure cooker for 10 min. The sections were then blocked in 2.5% normal horse serum for 20 min followed by primary antibody (CD63, 1:500, R&D Systems, #MAB50482) for 1 h. ImmPRESS horse reagent was added for 30 min, followed by DAB solution for 4 min. Slides were counterstained with hematoxylin, cleared from distilled water to xylene, and coverslipped. Imaging was performed on an AxioScan Z1 microscope. Image analysis was performed in ImageJ version 1.54d by RGB color thresholding. 

### 4.9. Statistical Analysis

Data are expressed as mean ± SD for in vitro studies and SEM for data generated from in vivo studies, unless otherwise noted. Statistical significance is abbreviated as * *p* < 0.05, ** *p* < 0.01, *** *p* < 0.001, and **** *p* < 0.0001. Sample sizes for each experiment can be found in the figure legends. The statistical analysis performed for immunocytochemistry staining was the Mann–Whitney two-tailed *t*-test. The statistical analysis performed for CD63 immunohistochemistry was the Kruskal–Wallis test. Statistics were performed in GraphPad Prism version 10.0.0.

## Figures and Tables

**Figure 1 ijms-25-02642-f001:**
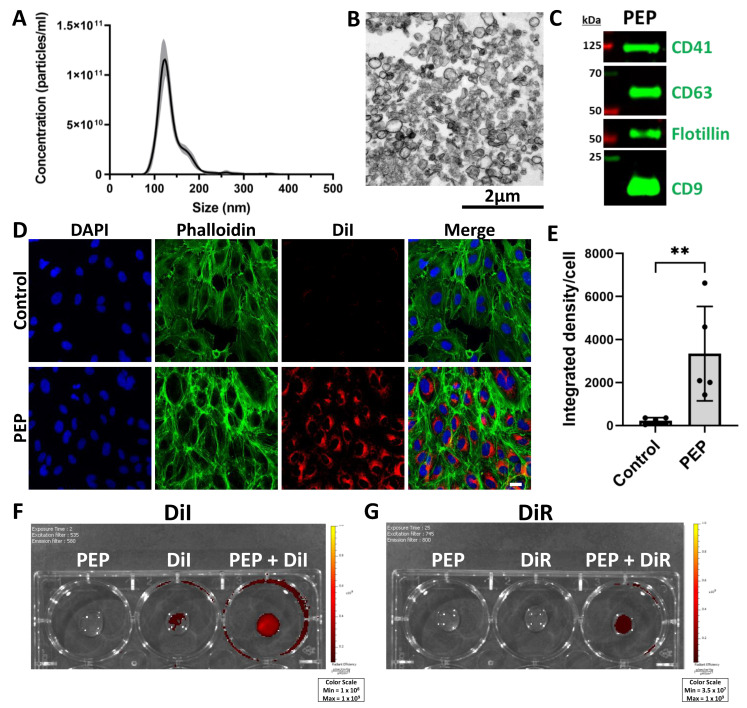
Exosome characterization of PEP and membrane labeling by lipophilic dye. (**A**) NanoSight analysis of PEP demonstrating distribution of particle size ± standard deviation (gray shading). (*n* = 3). (**B**) Transmission electron microscopy image of PEP with scale bar 2 µm. (**C**) Western blot demonstrating presence of hallmark exosomal proteins CD9, CD63, and Flotillin, as well as platelet integrin CD41. (**D**) Immunocytochemistry of human umbilical vein endothelial cells following treatment with DiI-labeled PEP or DiI alone (control) in red, counterstained with phalloidin (green) and DAPI (blue), scale bar 20 µm. (**E**) Quantification (*n* = 5) of mean integrated density DiI per nuclei (±SD) for Figure 1D. ** *p* < 0.001 using Mann-Whitney two-tailed *t*-test. (**F**) Xenogen image demonstrating fluorescent signal detected in unlabeled PEP, DiI, and PEP + DiI. (**G**) Xenogen image demonstrating fluorescent signal detected in unlabeled PEP, DiR, and PEP + DiR.

**Figure 2 ijms-25-02642-f002:**
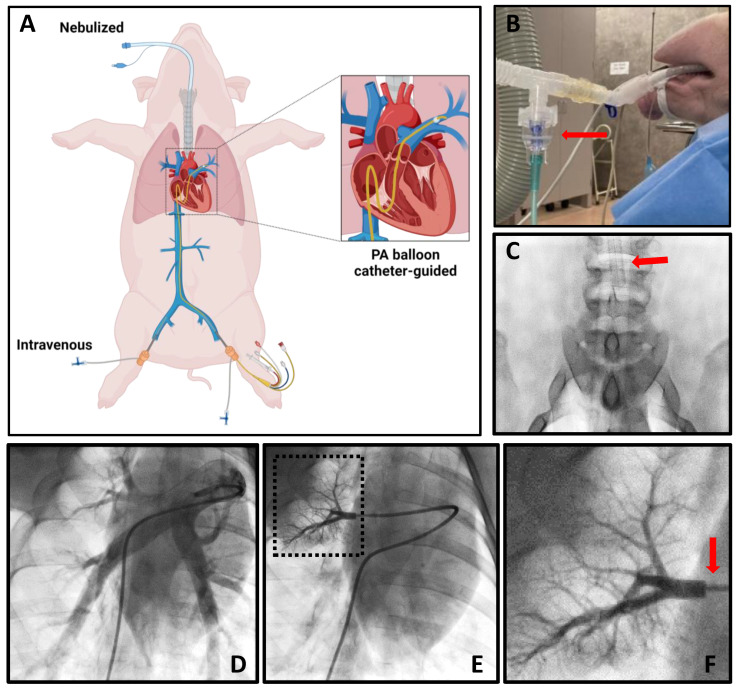
Pulmonary exosome delivery procedures. (**A**) Cartoon depicting methods employed for pulmonary delivery of exosomes, including nebulization, intravenous, and PA balloon catheter-guided approaches. (**B**) Photograph demonstrating jet nebulizer (red arrow) attachment to endotracheal tubing. (**C**) Fluoroscopy image showing placement of endotracheal tube (red arrow) in the trachea above the carina to allow bilateral pulmonary delivery. (**D**) Pulmonary angiogram of the right lung. (**E**) Angiogram demonstrating occlusion by balloon catheter of pulmonary artery branch with contrast injection and dashed box indicating area of zoomed in image in Figure 2F. (**F**) Zoomed-in image from Figure 2E with red arrow indicating area of balloon catheter occlusion.

**Figure 3 ijms-25-02642-f003:**
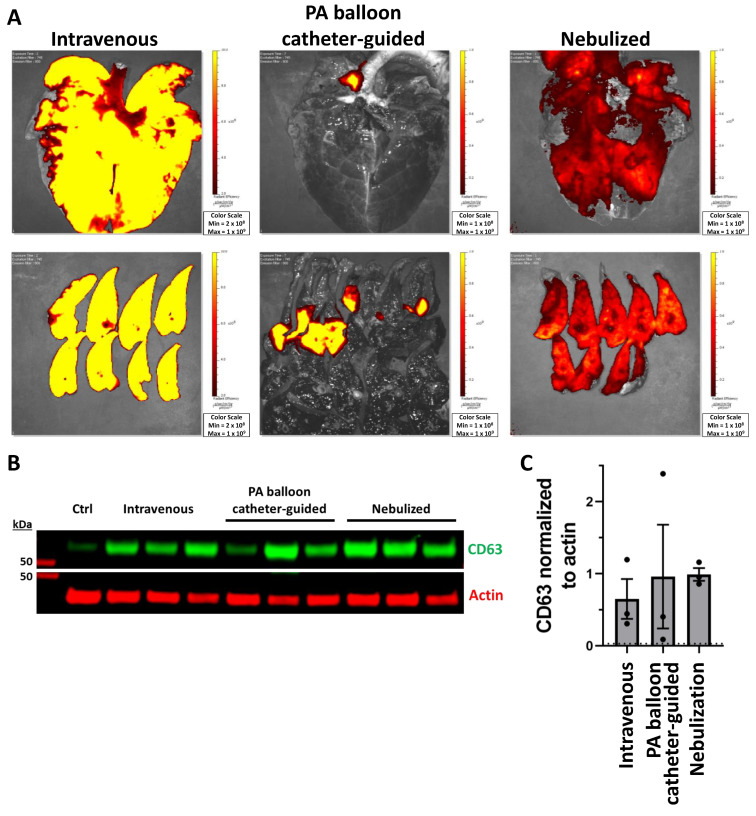
Pulmonary absorption of exosomes with intravenous, PA balloon catheter-guided, and nebulized delivery. (**A**) Xenogen imaging of uptake of DiR-labeled PEP in the lungs for intravenous, PA balloon catheter-guided, and nebulized delivery. (**B**) Western blot demonstrating presence of PEP in control lung (ctrl) tissue compared to lung tissue from intravenous, PA balloon catheter-guided, and nebulized deliveryusing exosomal protein CD63 (green) and loading control actin (red). (**C**) Quantification (*n* = 3) of mean fluorescent signal (±SEM) of CD63 normalized to actin loading control; dotted line represents level of control (ctrl).

**Figure 4 ijms-25-02642-f004:**
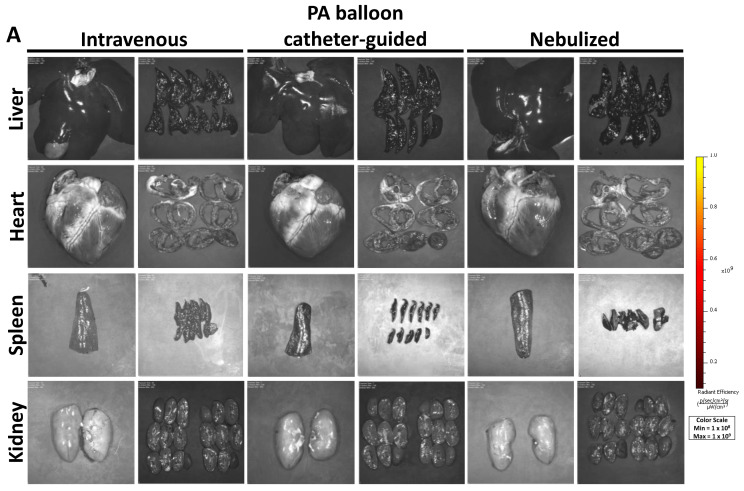
Off-target absorption of exosomes in the liver, heart, spleen, and kidney with intravenous, PA balloon catheter-guided, and nebulized delivery. (**A**) Xenogen imaging of uptake of DiR-labeled PEP in the heart, liver, spleen, and kidney with images of entire organ and organ sections shown. (**B**) Western blots demonstrating presence of PEP in liver tissue compared to PEP and control organ tissue (–ctrl) using exosomal protein CD63 (green) and loading control GAPDH (green). (**C**) Quantification (*n* = 3) of mean fluorescent signal (±SEM) of CD63 normalized to GAPDH loading control, with level of control liver tissue shown in dotted line. (**D**–**F**) Western blots demonstrating presence of PEP in heart (**D**), spleen (**E**), and kidney (**F**) compared to PEP and control organ tissue (–ctrl) using exosomal protein CD63 (green) and loading controls actin (red) or GAPDH (green).

**Figure 5 ijms-25-02642-f005:**
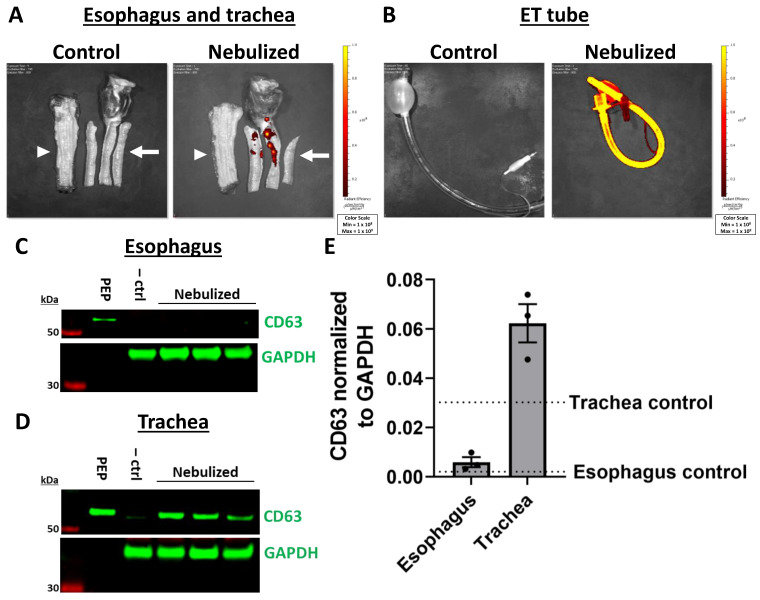
Off-target absorption of exosomes in the esophagus and trachea with intravenous, PA balloon catheter-guided, and nebulized delivery. (**A**) Xenogen imaging of DiR-labeled PEP uptake in the esophagus (white arrowhead) and trachea (white arrow) of control tissue compared to tissue exposed to nebulized PEP (top). (**B**) Endotracheal (ET) tube demonstrating loss of PEP on plastic tubing also shown (bottom). (**C**) Western blot demonstrating PEP uptake in the esophagus by nebulization compared to PEP and control pig esophagus tissue (–ctrl) with exosome marker CD63 (green) and GAPDH (green) loading control. (**D**) Western blot demonstrating PEP uptake in the trachea by nebulization compared to PEP and control pig trachea tissue (–ctrl) with exosome marker CD63 (green) and GAPDH (green) loading control. (**E**) Quantification (*n* = 3) of mean fluorescent signal (±SEM) of CD63 normalized to GAPDH loading control for trachea and esophagus with nebulization. The level of control esophagus or trachea tissue is shown with the labeled dotted lines.

**Figure 6 ijms-25-02642-f006:**
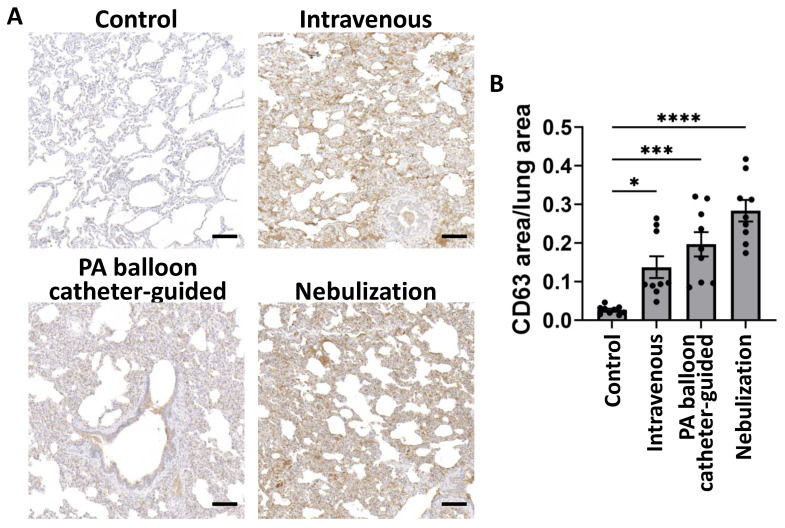
Histologic evaluation of pulmonary exosome delivery with intravenous, PA balloon catheter-guided, and nebulized delivery. (**A**) Immunohistochemical staining of lung tissue with CD63 (brown) with hematoxylin counterstain (blue) with scale bar 100 µm. (**B**) Mean (±SEM) ratio of CD63 area to total lung tissue area in untreated (control), intravenous, PA balloon catheter-guided, or nebulization delivery methods (*n* = 9). * *p* < 0.05, *** *p* < 0.001, and **** *p* < 0.0001 using Kruskal–Wallis test.

## Data Availability

All data described in this paper are shown in the main figures. Further information can be disclosed if desired by reasonable request to the corresponding author.

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
