# Peer review of "Pulmonary Biodistribution of Platelet-Derived Regenerative Exosomes in a Porcine Model"

_ijms, 2024, doi:10.3390/ijms25052642_

Round 1
Reviewer 1 Report
Comments and Suggestions for Authors
The manuscript “Pulmonary Biodistribution of Platelet-derived Regenerative Exosomes in a Porcine Model” by Dr. Atta Behfar et. Colleagues is a research article that investigates the biodistribution of exosomes in a live porcine animal model. In the manuscript, exosomes are delivered via different delivery methods, i.e. Nebulization, intravenously or even centrally, via a pulmonary artery balloon catheter. The manuscript aims to clarify which via of delivery is more suitable for exosome-based lung therapies. The innovation is limited as this has been done before in mice by other groups, but never in large animal models, not reducing the impact of the work.
Despite the initial enthusiasm for the work, unfortunately, the manuscript doesn’t follow the journal guidelines and it is not in the recommended template. The manuscript will be reviewed once the authors use the appropriate template, include images and figure descriptions within the text, and resubmit.
Author Response
We thank the reviewer for their comment and now have modified the manuscript to conform to the format that is required by the journal. We hope that in this modified form it is more suitable for review.
Reviewer 2 Report
Comments and Suggestions for Authors
This manuscript explores the pulmonary biodistribution of platelet-derived regenerative exosomes (PEP) in pigs using various delivery methods. Nebulization and intravenous infusion resulted in global lung uptake, while pulmonary artery balloon catheter-guided delivery localized exosomes. Nebulization showed significant liver uptake, possibly due to its longer administration time. The study emphasizes the effectiveness of nebulization for global pulmonary delivery and suggests balloon catheter-guided delivery for targeted approaches. These findings underscore the need for large animal models in exosome biodistribution studies and highlight the therapeutic potential of platelet-derived regenerative exosomes for pulmonary applications.
The manuscript would benefit from additional revisions:
1. Since there are no actual figures presented in the draft paper, it is impossible to assess the quality, adequacy, or accuracy of the presented results. It is essential to re-evaluate the paper once all figures are added.
2. In the "Animal Studies" section (4.6), please include additional information about whether the animal research was conducted in an accredited facility. Additionally, provide details such as genus and species (with the proper Latin designation), sex, internationally accepted genetic nomenclature, age, weight, and the source and origin of the animals used. Providing specific procedural details for these basic variables is crucial for enabling replication.
3. Include additional information on the control tissue for the biodistribution assay of platelet-derived regenerative exosomes (PEP) in lung tissue.
4. Clarify the rationale behind conducting only Western blot for CD63 in the biodistribution of PEP in off-target tissues, as opposed to the targeted lung tissue where a variety of markers such as CD63, Flotillin, CD9, and CD41 were used. Additionally, provide information on the control tissues used in these analyses.
Author Response
1 - We have now modified the manuscript to conform to the journals formatting such that the reviewer will be able to see the figures.
2 - The manuscript has now been modified on Page 10 lines 312-313 and Page 11 Lines 314-336 with the appropriate modifications.
3 - We appreciate the opportunity to highlight control tissue analysis. The method for control tissue analysis is described on Page 11 under sections 4.7 and 4.8. Furthermore, Figure 4 visualizes and quantitates by western blot, non-lung tissue samples.
4 - Halmark markers of platelet-derived exosomes include as the reviewer highlighted Tetraspanins CD9 and CD63 as well as platelet integrin marker CD41 and endosomal chaperone protein Flotillin. Although in vitro all the exosomes demonstrate expression of all of these markers, necessary to establish both phenotype and biogenesis, our team could only utilize the CD63 antibody that was capable of distinguishing Human versus Porcine tissue. The other markers had too much cross reactivity to accurately distinguish human platelet-derived exosomes in pig tissue. The discussion has now been modified to highlight this on page 9 lines 224-225. A description of control tissue is now provided in figure 4 and reviewed on page 6 lines 135-144.
Round 2
Reviewer 1 Report
Comments and Suggestions for Authors
Comments for the Manuscript “”Pulmonary Biodistribution of Platelet-derived regenerative Exosomes in a porcine model” by Dr. Atta Behfar. The aim of the study is to identify the optimal route of administration of platelet-derived exosomes to create new lung therapies. The authors have administered PEP via PA ballon, Nebulized or intravenously. The methods for PEP quantification are not optimal, but a lot of merit is due to the complicated surgical procedures and catheterization performed. Besides a few other manuscripts have focused on EV distribution in mice and rats, this work is innovative as the porcine investigation can provide data of higher translatability into clinical research. There are some concerns related to the design of the study. In the manuscript the is no mention of the reason for choosing platelet-derived exosomes. Endothelial cells have shown the ability to internalize EVs both autologous (endothelial-derived) and heterologous (produced by other cell population). However, the absorbance of heterologous EVs is lower, thus suggesting that the results proposed here may vary when using EV-derived from different cell populations. Thus, A major comment:
1. The abstract, introduction, and discussion need to be modified to include the reason for choosing exosomes derived from platelets. Authors should clearly state, so that a general audience will be able to understand, the reasons behind, the rationale for platelet-derived products, and evidence from the literature for their beneficial effects in lung therapies (or also other applications). Different authors found that platelet derived vesicles promote cancer differentiation (Janowska-Wieczorek A, Wysoczynski M, Kijowski J, Marquez-Curtis L, Machalinski B, Ratajczak J. et al. Microvesicles derived from activated platelets induce metastasis and angiogenesis in lung cancer. Int J Cancer. 2005;113:752–60; Pucci F, Rickelt S, Newton AP, Garris C, Nunes E, Evavold C. et al. PF4 Promotes Platelet Production and Lung Cancer Growth. Cell Rep. 2016;17:1764–72.) how this evidence fits with the purpose of the authors’s work? If it doesn’t include this in the limitations of the work. Additional discussion is required on the autologous vs heterologous EV administration and how these results may vary when utilizing products derived from other cells.
Author Response
Thank you for this point, we appreciate the opportunity to improve the clarity for our choice in exosomes. We have added additional information in the abstract (line 15), introduction (lines 56-63), and discussion (lines 225-226) clarifying the choice in using platelet-derived exosomes. Our laboratory has previously shown these exosomes to be effective in inducing wound healing and tissue regeneration, and we have added citations reflecting this work. With regards to platelet extracellular vesicles in lung cancer, previous in vivo studies by our group have not documented any cancer formation in response to platelet-derived exosomes. The findings of the cited studies appear to be specific to platelet extracellular vesicle impact on a pre-existing lung cancer, while our studies we focus on regeneration in the setting of tissue injury. We have also added discussion of autologous vs heterologous exosomes (lines 270-277).
Reviewer 2 Report
Comments and Suggestions for Authors
Author's Notes
1 - We have now modified the manuscript to conform to the journals formatting such that the reviewer will be able to see the figures.
Reply: Please add a molecular ladder to Figures 1, 3, 4, and 5 for comparison, preferably presented on the same blot. At present, it appears that all bands have been cut out of the picture, making it difficult to adequately demonstrate the presence of target proteins. It is strongly advised to include quantitative comparisons between samples on the same gels/blots, with clearly indicated rearranged lanes, and cropped gels retaining all significant bands.
Additionally, ensure that loading controls (e.g., GADPH, actin) are run on the same blot for accurate assessment. This will enhance the reliability and credibility of the presented data.
2 - The manuscript has now been modified on Page 10 lines 312-313 and Page 11 Lines 314-336 with the appropriate modifications.
Reply: There are no such lines on the abovementioned pages. Please specify again.
3 - We appreciate the opportunity to highlight control tissue analysis. The method for control tissue analysis is described on Page 11 under sections 4.7 and 4.8. Furthermore, Figure 4 visualizes and quantitates by western blot, non-lung tissue samples.
Reply: Revisions are accepted.
4 - Halmark markers of platelet-derived exosomes include as the reviewer highlighted Tetraspanins CD9 and CD63 as well as platelet integrin marker CD41 and endosomal chaperone protein Flotillin. Although in vitro all the exosomes demonstrate expression of all of these markers, necessary to establish both phenotype and biogenesis, our team could only utilize the CD63 antibody that was capable of distinguishing Human versus Porcine tissue. The other markers had too much cross reactivity to accurately distinguish human platelet-derived exosomes in pig tissue. The discussion has now been modified to highlight this on page 9 lines 224-225. A description of control tissue is now provided in figure 4 and reviewed on page 6 lines 135-144.
Reply: Revisions are accepted.
Author Response
1 - We have now modified the manuscript to conform to the journals formatting such that the reviewer will be able to see the figures.
Reply: Please add a molecular ladder to Figures 1, 3, 4, and 5 for comparison, preferably presented on the same blot. At present, it appears that all bands have been cut out of the picture, making it difficult to adequately demonstrate the presence of target proteins. It is strongly advised to include quantitative comparisons between samples on the same gels/blots, with clearly indicated rearranged lanes, and cropped gels retaining all significant bands.
Additionally, ensure that loading controls (e.g., GADPH, actin) are run on the same blot for accurate assessment. This will enhance the reliability and credibility of the presented data.
AUTHOR REPLY:
Western blots in figures 1, 3, 4, and 5 have been adjusted to include the molecular ladder from the blot. All quantitation performed was between samples on the same gel/blot. Loading controls are also run on the same blots as the target proteins.
2 - The manuscript has now been modified on Page 10 lines 312-313 and Page 11 Lines 314-336 with the appropriate modifications.
Reply: There are no such lines on the abovementioned pages. Please specify again.
AUTHOR REPLY:
Our apologies, please see section 4.6 “Animal studies” for the requested information on animal research including genus, species, sex, genetic nomenclature, age, weight, and source of animals used.
3 - We appreciate the opportunity to highlight control tissue analysis. The method for control tissue analysis is described on Page 11 under sections 4.7 and 4.8. Furthermore, Figure 4 visualizes and quantitates by western blot, non-lung tissue samples.
Reply: Revisions are accepted.
We appreciate your comment.
4 - Halmark markers of platelet-derived exosomes include as the reviewer highlighted Tetraspanins CD9 and CD63 as well as platelet integrin marker CD41 and endosomal chaperone protein Flotillin. Although in vitro all the exosomes demonstrate expression of all of these markers, necessary to establish both phenotype and biogenesis, our team could only utilize the CD63 antibody that was capable of distinguishing Human versus Porcine tissue. The other markers had too much cross reactivity to accurately distinguish human platelet-derived exosomes in pig tissue. The discussion has now been modified to highlight this on page 9 lines 224-225. A description of control tissue is now provided in figure 4 and reviewed on page 6 lines 135-144.
Reply: Revisions are accepted.
We thank the reviewer for their comment.